# Comparison of the Biochemical Properties and Roles in the Xyloglucan-Rich Biomass Degradation of a GH74 Xyloglucanase and Its CBM-Deleted Variant from *Thielavia terrestris*

**DOI:** 10.3390/ijms23095276

**Published:** 2022-05-09

**Authors:** Beibei Wang, Kaixiang Chen, Peiyu Zhang, Liangkun Long, Shaojun Ding

**Affiliations:** Jiangsu Co-Innovation Center of Efficient Processing and Utilization of Forest Resources, College of Chemical Engineering, Nanjing Forestry University, Nanjing 210037, China; jason22@njfu.edu.cn (B.W.); chenkaixiang@njfu.edu.cn (K.C.); zhangpeiyu@njfu.edu.cn (P.Z.); longlk602@njfu.edu.cn (L.L.)

**Keywords:** *Thielavia terrestris*, Xyloglucan, glycoside hydrolase family 74 (GH74) xyloglucanase, family 1 carbohydrate-binding module (CBM1), synergism, lignocellulosic biomass

## Abstract

Xyloglucan is closely associated with cellulose and still retained with some modification in pretreated lignocellulose; however, its influence on lignocellulose biodegradation is less understood. *Tt*GH74 from *Thielavia terrestri*s displayed much higher catalytic activity than previously characterized fungal GH74 xyloglucanases. The carbohydrate-binding module 1 (CBM1) deleted variant (*Tt*GH74ΔCBM) had the same optimum temperature and pH but an elevated thermostability. *Tt*GH74 displayed a high binding affinity on xyloglucan and cellulose, while *Tt*GH74ΔCBM completely lost the adsorption capability on cellulose. Their hydrolysis action alone or in combination with other glycoside hydrolases on the free xyloglucan, xyloglucan-coated phosphoric acid-swollen cellulose or pretreated corn bran and apple pomace was compared. CBM1 might not be essential for the hydrolysis of free xyloglucan but still effective for the associated xyloglucan to an extent. *Tt*GH74 alone or synergistically acting with the CBH1/EG1 mixture was more effective in the hydrolysis of xyloglucan in corn bran, while *Tt*GH74ΔCBM showed relatively higher catalytic activity on apple pomace, indicating that the role and significance of CBM1 are substrate-specific. The degrees of synergy for *Tt*GH74 or *Tt*GH74ΔCBM with the CBH1/EG1 mixture reached 1.22–2.02. The addition of GH10 xylanase in *Tt*GH74 or the *Tt*GH74ΔCBM/CBH1/EG1 mixture further improved the overall hydrolysis efficiency, and the degrees of synergy were up to 1.50–2.16.

## 1. Introduction

Lignocellulosic biomass has become a potential feedstock for biofuels and biochemicals production [1], and this mainly consists of cellulose (25–50%), hemicellulose (15–35%) and lignin (5–30%) with small amounts of pectin and wall proteins [2]. Cellulose, the principal component of lignocellulosic biomass, exists as long unbranched fibers composed of β-D-(1,4)-linked glucan. The cellulose microfibers in lignocellulose are embedded in a matrix composed of hemicellulose, acidic pectin polysaccharide, structural glycoprotein and lignin [3,4]. 

Hemicelluloses are closely associated with the cellulose fibrils and lignin, which limits the accessibility of cellulose to cellulase, hence, greatly hindering the hydrolysis efficiency of cellulase. Hemicelluloses can be divided into four groups, xylan, mannan, xyloglucan and mixed linkage β-glucan [5]. Among them, xyloglucan (XG) is a major complex polysaccharide in the primary cell walls of most higher plants, and it is closely associated with cellulose and covers part of the cellulose surface [6,7].

In addition, XG not only has hydrogen bonds with cellulose but also has ester bonds to the—COOH groups of pectin and benzyl-sugar ether bonds to ferulic acid [7]. XG has a β-1,4-glucan backbone that is highly branched with α-xylose residues linked to glucose at the O-6 position. The side chain xylosyl residues can be further substituted with different monosaccharides, disaccharides or trisaccharides. 

A nomenclature for xyloglucan-derived oligosaccharides was first introduced by Fry et al. [8]. The letter G indicates an unbranched glucose residue, and the X represents xylose linked to glucose backbone (α-D-xylose-1,6-β-D-glucose). The xylose residues can carry a β-D-galactose (L motif, β-D-galactose-1,2-α-D-xylose-1,6-β-D-glucose), while galactose residues in L can carry an α-L-fucose (F motif, α-L-fucose-1,2-β-D-galactose-1,2-α-D-xylose-1,6-β-D-glucose), etc. [9].

The bioconversion of lignocellulosic biomass into biofuels and biochemicals needs three steps: (1) pretreatment to improve the accessibility of lignocellulose, (2) enzymatic saccharification to produce fermentable sugars and (3) microbial fermentation to obtain the target products [10]. The hydrolytic efficiency and cost of enzymes in the second step are major factors restricting the cost-effective production of biofuels and biochemicals from lignocellulose due to biomass recalcitrance [11].

In view of the complex and strong network structure formed by XG with other lignocellulosic components, we presumed that XG is very difficult to remove during common pretreatment and may remain in the pretreated residues, particularly in some XG-rich biomasses, such as corn bran and apple pomace, therefore, limiting the accessibility and hydrolysis efficiency of cellulose [12,13]. The removal of XG may be essential to promote the enzymatic hydrolysis of pretreated lignocelluloses.

Microbial xyloglucanase was first discovered from the *Aspergillus aculeatus* in 1999 [14]. Xyloglucanases have been classified into glycoside hydrolase (GH) families 5, 9, 12, 16, 44, 45 and 74 in the Carbohydrate-Active Enzymes (CAZy, http://www.cazy.org/; accessed on 9 March 2022) database [15]. Most of the xyloglucanases from fungi belong to GH12 and GH74 [16]. Of these, GH74 xyloglucanases are known to have a high specific activity towards XG. GH74 xyloglucanases consist of two seven-bladed β-propeller domains that form a large interfacial cleft to accommodate the bulky polysaccharide [17]. 

GH74 xyloglucanases have been identified and divided into three modes of activity: *Exo*, *endo*-dissociative and *endo*-processive. *Exo*-xyloglucanases recognize the reducing ends of XG and release two glycosyl residue segments. Both *endo*-xyloglucanases hydrolyze the internal β-1,4-glucan backbone of XG in the initial stage. The difference is that *endo*-processive-xyloglucanases can progressively hydrolyze the XG chains before desorption. In contrast, *endo*-dissociative-xyloglucanases hydrolyze the backbone of XG and release both new chain ends, subsequently, resulting in desorption from the XG polysaccharide [17,18].

The carbohydrate-binding modules (CBMs) are non-catalytic modules [19], which have several global roles in functionality, including coordinated glycan recognition, general substrate adherence and structure–function contributions to the catalytic site [20]. According to the CAZy database, about 60% of GH74 gene structures from fungi contain family 1 carbohydrate-binding module (CBM1). 

CBM1 belongs to the surface binding CBM group (type A) [21], which can interact with insoluble and crystalline polysaccharides and target the enzymes to the substrate surface, therefore, enhancing the enzymatic activity [22,23]. At present, there is little literature concerning the effects of CBM1 on GH74 xyloglucanases functionality. It is unclear how CBM1 affects the substrate binding and enzymatic hydrolysis efficiency of GH74 xyloglucanases when degrading XG in lignocellulose. 

Xyloglucanases are widely distributed in bacteria and fungi, and they are usually co-expressed with cellulase and xylanase [24,25,26,27] indicating that they may have synergy with cellulase and xylanase for the efficient hydrolysis of lignocellulose biomass. Recently, it was reported that the deletion of xyloglucanase gene (*cel74a*) significantly affected the expression of holocellulolytic genes and reduced the degradation efficiency of sugarcane bagasse [28].

Previous experiments showed that added the so-called accessory enzyme xyloglucanase to the cellulase mixture, the hydrolytic performance of the cellulase mixture on pretreated biomass (including corn stover, corn fiber, sweet sorghum bagasse, barley straw, reed canary grass and willow) could be improved [29,30]. However, compared to the extensive studies of the influence of xylan and mannan in the lignocellulose hydrolysis [31,32,33,34], the effects of XG on the overall lignocellulose hydrolysis efficiency is still less understood, and the role of xyloglucanases needs to be elucidated in depth.

*Thielavia terrestris* is a thermophilic filamentous fungus and can grow normally at 40–50 °C. It contains a variety of heat-resistant cellulose/hemicellulose hydrolases and can hydrolyze a variety of major polysaccharides in biomass [35]. This strain is a potential source of enzymes with scientific and commercial interests [36]. In this study, one GH74 xyloglucanase from *T. terrestris* (*Tt*GH74) and its CBM1-deleted variant (*Tt*GH74ΔCBM) are expressed in *Pichia pastoris,* and their characteristics are comprehensively compared.

In addition, their hydrolytic action alone or in combination with cellulase and xylanase on free-XG, xyloglucan-coated phosphoric acid-swollen cellulose (PASC) and different pretreated natural xyloglucan-rich biomasses (corn bran and apple pomace) were also investigated with the aims to deepen our understanding of the effects of the associated XG on the overall hydrolysis efficiency and the role of GH74 xyloglucanases in the degradation of XG-rich biomasses.

## 2. Results and Discussion

### 2.1. Expression and Purification of TtGH74 and TtGH74ΔCBM

The structural diagram of *Tt*GH74 and *Tt*GH74ΔCBM is shown in Figure 1a. After codon optimization, the codon adaptation index of the *Tt*GH74 was 0.98, and the percentage of GC content was 40.93%. The codon-optimized genes encoding GH74 xyloglucanase from *T. terrestris* (*Tt*GH74) and CBM-deleted variant *Tt*GH74ΔCBM were successfully expressed in *P. pastoris* KM71H. The purified enzymes were analyzed by SDS-PAGE, and the molecular masses of *Tt*GH74 and *Tt*GH74ΔCBM were about 90 and 85 kDa, respectively. After digestion by N-glycosidase Endo H, SDS-PAGE analysis showed that their molecular masses decreased closely to theoretical values of 87.89 and 83.09 kDa, respectively (Figure 1b).

The analysis of the amino acid sequences of *Tt*GH74 and *Tt*GH74ΔCBM by NetNGlyc 1.0 Server (https://services.healthtech.dtu.dk/service.php?NetNGlyc-1.0; accessed on 9 March 2022) confirmed that they have three putative N-glycosylation sites (N212, N325 and N409), thus, indicating that the recombinant *Tt*GH74 and *Tt*GH74ΔCBM could partially be N-glycosylated.

The protein expression level of *Tt*GH74 was only 60% of that of *Tt*GH74ΔCBM, therefore, the expression conditions of *Tt*GH74 were optimized. The methanol-regulated promoter of the alcohol oxidase 1 gene (AOX1) is the key to the high expression level of recombinant protein in *P. pastoris* [37], which was significantly influenced by the methanol concentration. In addition, the expression level was also affected by the medium’s pH, induction temperature and time. Under the optimized condition, the enzyme activity of *Tt*GH74 secreted in the supernatant reached 94.65 U/mL after 5 days of induction with 1.5% methanol at pH 6.0 and 28 °C (Appendix A).

### 2.2. Properties of Recombinant TtGH74 and TtGH74ΔCBM from P. pastoris

Among all tested substrates, *Tt*GH74 showed ultra-high activity against tamarind seed XG. *Tt*GH74 also showed activities toward barley β-glucan, konjac glucomannan, lichenan and PASC; however, their corresponding activities were only approximately 5% and 1% of that on XG. No activities were detected towards other polysaccharides, such as laminarin, starch, xylan, pectin and chitin. (Table 1).

*Tt*GH74 and *Tt*GH74ΔCBM displayed the maximum activity toward tamarind seed XG at 75 °C and pH 5.5 (Figure 2a,b). However, the enzymes were unstable at 70 °C. After pre-incubation at 70 °C for 30 min, they completely lost their activities (Figure 2c). This result indicated that the enzymes had more thermostability in the presence of the substrate, which may be attributed to the stabilizing effects of the substrate [38]. After pre-incubation at 65 °C for 12 h, both enzymes lost their activities completely. Notably, the deletion of CBM1 significantly improved the thermostability of *Tt*GH74ΔCBM when the pre-incubation of enzymes was at temperatures below 65 °C. 

With the pre-incubation of enzymes at 60 °C to 55 °C for 12 h, the residual activity of *Tt*GH74ΔCBM was 6% to 11% higher than that of *Tt*GH74. With the pre-incubation of enzymes at 50 °C for 12 h, the activity of *Tt*GH74 remained above 77%, while the activity of *Tt*GH74ΔCBM remained above 89% (Figure 2c). The data differences regarding *Tt*GH74 and *Tt*GH74ΔCBM thermostability were statistically significant (*p* < 0.05). Conversely, the thermostability of some glycoside hydrolases from fungi was decreased after the removal of CBM1 [39,40]. The inconsistent observation about CBM1 suggests that its role in thermostability might be related to the kinds of enzymes. The enzymes were stable in a pH range from 3 to 8 (Figure 2d).

The effects of metal ions (1 and 5 mM) on the enzyme activities was investigated, and the results are shown in Appendix A. The activities of *Tt*GH74 and *Tt*GH74ΔCBM were increased by 2–25% in the presence of 5 mM K^+^, Ca^2+^, Ba^2+^ and Ni^2+^ ions. At the same time, the enzyme activities were decreased by 5–75% in the presence of (1 and 5 mM) EDTA, NH_4_^+^, Li^+^, Mg^2+^, Pb^2+^, Zn^2+^, Cu^2+^, Mn^2+^, Co^2+^, Fe^3+^ and Al^3+^. We found that 5 mM Pb^2+^ and Fe^3+^ could dramatically reduce the activities of *Tt*GH74 and *Tt*GH74ΔCBM by 90%.

The kinetic values were determined in the concentration ranges of 0.2–6.0 mg/mL of tamarind seed XG (Table 2). It can be seen from Table 2 that the catalytic activity of *Tt*GH74 is much higher than the previously characterized GH74 xyloglucanases. *Tt*GH74ΔCBM has a slightly smaller *K*_m_ and *K*_cat_ value than *Tt*GH74 (Table 2). Research reported that the *K*_cat_ of GH74 xyloglucanase from *Phanerochaete chrysosporium* was increased slightly after the deletion of CBM1 [41]. These results suggested that CBM1 in GH74 xyloglucanases slightly affected its catalytic activity for the soluble XG.

The end-products generated from the tamarind seed XG after *Tt*GH74 hydrolysis were identified using matrix assisted laser desorption/ionization time-of-flight mass spectrometry (MALDI-TOF MS). A series of sodium adducts of the hydrolyzed products were detected (Figure 3). The m/z values of the major peaks were 1085, 1247 and 1409 corresponding to molecular masses of the reduced oligosaccharide ions [XXXG + Na]^+^, [XXLG + Na]^+^ and [XLLG + Na]^+^, respectively. 

Apart from the main XG building blocks, various low molecular products were generated during the hydrolysis and the m/z values detected by mass spectrometry may potentially correspond to XX, XXG, GXX, XGX, XL, XLG, GXL, LL, LG, XXL, LLG, GXXXG, GXLLG and XLLGX oligosaccharides, respectively; however, no X, G and L were detected. Although *Tt*GH74 randomly hydrolyzed the XG backbone, it did not cleave at the reducing end side of L units in XLXG and XLLG [46] as shown in the schematic diagram. 

*Tt*GH74 released oligo-xyloglucans, such as XXXG, XXLG and XLLG, which were the typical final products of *endo*-type xyloglucanases. In addition, the sequence alignments of *Tt*GH74 with other fungal GH74 xyloglucanases (Appendix A) revealed that *Tt*GH74 has four conserved tryptophan residues that were regarded as the key amino acid residues for the *endo*-processive activity of GH74 xyloglucanases [17]. Thus, this indicates that *Tt*GH74 is an *endo*-processive-type xyloglucanase.

### 2.3. Adsorption of TtGH74 and TtGH74ΔCBM on Different Substrates

The binding affinity of *Tt*GH74 and *Tt*GH74ΔCBM on different substrates (10 mg in 0.5 mL) was compared by measuring the unbound protein in the supernatant. The results are shown in Figure 4a. All *Tt*GH74 protein bound onto PASC, while only approximately 35% and 23% of *Tt*GH74 bound onto Avicel and Whatman filter paper, respectively. For pretreated lignocellulosic substrates, the amounts of bound protein in ascending order are 15%, 20% and 30% for deep eutectic solvents (DES)-pretreated corn directly proportional to the galactose content in the substrates (Table 3). 

Without CBM1, the adsorption capacity of *Tt*GH74ΔCBM for insoluble carbohydrates is almost completely lost. We further measured the adsorption of *Tt*GH74 under the condition of low cellulose content (1 mg in 0.5 mL) with additional different proportions of XG (Figure 4b). Due to the relative excess amount of enzyme, the unbound protein was detected in supernatant; however, with the increase of XG proportion, the amount of the unbound protein gradually decreased, which implied that the *Tt*GH74 could bind onto XG. To further verify the adsorption of CBM1 on XG, we compared the mobility of the CBM-deleted variant *Tt*GH74ΔCBM with the *Tt*GH74 in xyloglucan-containing native gels. 

We observed that the mobility of *Tt*GH74ΔCBM increased significantly compared to *Tt*GH74, which further clarified the binding affinity of CBM1 in *Tt*GH74 onto XG (Figure 4c,d). The binding affinity of CBM1 on cellulose and the role of CBM1 in cellulase functionality were well demonstrated in the literature [47,48]. CBM1 is type A CBM and was previously recognized as a cellulose-binding domain due to the first discovery in fungal cellulases [49]. However, the effects of CBM1 on XG binding and the activity of fungal GH74 xyloglucanases have been little reported. Our results suggested that CBM1 in *Tt*GH74 displayed a high binding affinity for both cellulose and XG.

### 2.4. Hydrolysis Action of TtGH74 and TtGH74ΔCBM on XG and XG-Coated PASC

Hydrolysis action of *Tt*GH74 and *Tt*GH74ΔCBM on free XG was performed in the concentration ranges of 25–600 μg XG (Figure 5a). When the amount of XG was low (less than 250 μg), *Tt*GH74ΔCBM exhibited a slightly lower hydrolysis yield than the intact enzyme due to its low affinity to the substrate. However, as the amount of XG increased, the yield of reducing sugars by *Tt*GH74ΔCBM hydrolysis increased significantly and surpassed the yield by *Tt*GH74 as the content of XG was over 250 μg.

XG-coated PASC with different proportions of XG were used as the substrates to investigate the influence of CBM1 in enzyme functionality. XG can coat onto the surface of PASC once mixture, however, the association pattern between XG with PASC relies on the proportion ratio of XG/PASC. The previous research revealed that at a low XG/cellulose concentration ratio (25 μg/mg), all XG is tightly bound to the cellulose surface, and XG is not easy to be hydrolyzed by the enzyme. However, with the increase of XG/cellulose ratio, XG forms accessible “loops” and “tails” on the cellulose surface, and the accessible XG gradually increases to a constant value [50]. The schematic diagram of the association pattern of XG and PASC is shown in Figure 5b. In addition, when the XG/PASC ratio was over 400 μg/mg, free XG could be detected in the liquid fraction (Appendix A).

When XG-coated PASC was reacted with *Tt*GH74 or *Tt*GH74ΔCBM (Figure 5a), XG was more difficult to be hydrolyzed than free XG by *Tt*GH74 and *Tt*GH74ΔCBM because of the close association of XG with PASC. Similar as free XG, once the proportion of XG increased over 400 μg/mg, *Tt*GH74ΔCBM released more reducing sugars from XG-coated PASC than *Tt*GH74. Since free XG was detected in the supernatant of XG-coated PASC solution with a high XG/cellulose ratio; thus, the higher catalytic performance of *Tt*GH74ΔCBM than *Tt*GH74 in the hydrolysis of XG-coated PASC solution as the proportion of XG over 400 μg/mg might be partially due to the presence of free XG in solution. 

However, relatively higher reducing sugars were released from XG-coated PASC by *Tt*GH74 than from *Tt*GH74ΔCBM as the proportional ratio of XG/PASC was below 300 μg/mg. These results indicated that CBM1 might not be essential for the hydrolysis of free XG but was effective for the associated XG to some degree. The coverage of XG on PASC even at low concentrations resulted in a significant decrease in the catalytic activity of endoglucanase 1 (EG1); in comparison, a smaller reduction in cellobiohydrolase 1 (CBH1) activity was observed (Figure 5a). 

These differences might be attributed to the different catalytic patterns between EG1 and CBH1. EG1 is a GH5 *endo*-glucanase, which randomly attacks on internal sites in the cellulose chain [51], while CBH1 is an *exo*-cellulase, which processively hydrolyzes the cellulose chain from reducing end [52].

### 2.5. Evaluation of the Presence of XG and Hydrolysis Action of TtGH74 and TtGH74ΔCBM on Pretreated Corn Bran and Apple Pomace

To evaluate the presence of XG in solid residues after pretreatment in corn bran and apple pomace, two substrates were pretreated at 90 °C by 1% sulfuric acid or DES at different times (3, 6, 9 and 12 h). The residues were then hydrolyzed by *Tt*GH74 or *Tt*GH74ΔCBM. As shown in Figure 6, a large amount of reducing sugars was released from all pretreated residues, indicating that XG was still retained after pretreatment. Interestingly, no matter what kinds of pretreatment methods and times used, *Tt*GH74 produced more reducing sugars than *Tt*GH74ΔCBM from pretreated corn bran after four days of reaction, while *Tt*GH74ΔCBM had better hydrolysis performance on DES-pretreated apple pomace than *Tt*GH74 (Figure 6). 

Finally, corn bran pretreated with sulfuric acid for 6 h, corn bran and apple pomace pretreated with DES for 9 h were used as substrates for further exploration. As shown in Table 3, three pretreated residues differed in chemical compositions due to different sources and pretreatment methods. DES pretreatment was more effective in the removal of lignin and hemicelluloses than sulfuric-acid pretreatment for corn bran (*p* < 0.01). Thus, in this study, only DES was used to pretreat apple pomace. The DES-pretreated apple pomace had the highest content of galactose among the three substrates (*p* < 0.01). Galactose is a specific substituent in the side branch of XG. 

Galactose may also come from the main chain of residual pectin; however, galactose in the pectin of apple accounts for approximately 1% of total pectin according to the literature [53], and thus the galactose in pretreated apple pomace mainly come from the side chain of XG. Therefore, the higher content of galactose represented the higher amount of XG. This might explain the higher adsorption capacity of *Tt*GH74 on DES-pretreated apple pomace than DES-pretreated and sulfuric-acid-pretreated corn bran (Figure 4a). 

This might also explain the much higher reducing sugars released from DES-pretreated apple pomace by either *Tt*GH74 or *Tt*GH74ΔCBM (Figure 6c). The content of lignin in DES-pretreated corn bran is much lower than sulfuric-acid-pretreated corn bran (Table 3). This might contribute to the more reducing sugars generated from DES-pretreated corn bran than sulfuric-acid-pretreated corn bran, although the content of XG is relatively lower in DES-pretreated corn bran.

The presence of XG in three pretreated substrates was confirmed by detecting the *Tt*GH74 hydrolysis end-products using high performance anion exchange chromatography with pulsed amperometric detection (HPAEC-PAD), and the formation of oligosaccharides produced by *Tt*GH74 hydrolysis of three pretreated substrates were further confirmed by MALDI-TOF MS.

Comparing Figure 7a,c, the profiles of the peak of oligosaccharides produced by *Tt*GH74 hydrolysis of corn bran with different pretreatment methods were similar in HPAEC-PAD analysis; however, the size of the peaks were different (the special peaks were numbered according to the elution time). For example, the proportion of peak 4 in Figure 7c was significantly higher than that in Figure 7a. However, for different substrates with the same pretreatment method, the profiles of the overall peak were different but the proportion of peak 4 was the same (Figure 7c,e).

In the MALDI-TOF MS spectrometry (Figure 7b,d,f), some detected m/z values might correspond to XG, XX, LG, XXG, XGX, XL, GGL, XLG, GXL, XXL, XXXG, LLG, XXLG and GXXXG oligosaccharides, respectively. A more amount of GGL, XLG, GXL and LLG oligosaccharides were released from sulfuric-acid-pretreated corn bran than that released from DES-pretreated corn bran. While more XG, XX, XL, XXG and XGX oligosaccharides were released from DES-pretreated corn bran than sulfuric-acid-pretreated corn bran (Figure 7b,d).

It could be inferred that xyloglucan in sulfuric-acid-pretreated corn bran contains more and longer branch chains than that in DES-pretreated corn bran. Like DES pretreated corn bran, more oligosaccharides of XG, XX, XL, XXG and XGX were released from DES-pretreated apple pomace (Figure 7d,f); however, many XLG and GXL were also released from DES-pretreated apple pomace, which was due to the difference between the composition of xyloglucan in apple pomace and corn bran.

In summary, different pretreatment methods might have various fractionalization and modification effects on xyloglucan in lignocellulosic biomass, resulting in the content change and structural modification of xyloglucan in different pretreated lignocellulosic biomasses; therefore, the end-product profiles in hydrolysates are different.

The time courses of enzymatic hydrolysis of three pretreated substrates in different concentrations (from 10 to 80 mg) were performed, and the results are shown in Figure 8. In general, *Tt*GH74 produced more reducing sugars than *Tt*GH74ΔCBM from sulfuric-acid-pretreated corn bran and DES-pretreated corn bran after four days of hydrolysis; however, the increase ranges varied depending on the concentration of substrates. For the two pretreated corn brans, at low substrate concentration (10 mg), the yields of reducing sugar by *Tt*GH74 were 1.23 (*p* < 0.05) and 1.39 (*p* < 0.05)-times higher than *Tt*GH74ΔCBM. The yields of reducing sugar by *Tt*GH74 were 1.36 (*p* < 0.05) and 1.13 (*p* < 0.05)-times higher than *Tt*GH74ΔCBM when the substrate concentration was 20 mg. 

At concentrations of 40–80 mg, the yield of reducing sugar by *Tt*GH74 was about 1.12 (*p* < 0.05)-times higher than that of *Tt*GH74ΔCBM. On the contrary, *Tt*GH74ΔCBM produced more reducing sugars than *Tt*GH74 from DES-pretreated apple pomace, no matter what concentration of substrate was used. The yields of reducing sugar by *Tt*GH74ΔCBM were 1.68–2.06 (*p* < 0.03)-times higher than that of *Tt*GH74 at concentrations of 10–80 mg. From the time course experiment, we concluded that the presence of CBM1 is conducive to enzymatic hydrolysis; however, its role and significance are substrate-specific because of the differences in the contents and structure of XG in different biomasses.

### 2.6. Synergistic Action of TtGH74 or TtGH74ΔCBM with CBH1/EG1 Mixture and Xylanase on Pretreated Corn Bran and Apple Pomace

As specific xyloglucan degrading enzymes, GH74 xyloglucanases were widely distributed in bacteria and fungi, and they were usually co-expressed with cellulase and xylanase. In view of the presence of XG in pretreated corn bran and apple pomace (Table 3) and its blocking effect on EG1 and CBH1 activity towards XG-coated PASC (Figure 5), it is reasonable to believe that xyloglucanases may be necessary for the efficient enzymatic saccharification of XG-rich biomasses. 

Thus, we further investigated the synergy between xyloglucanase with xylanase and cellulase in the hydrolysis of pretreated corn bran and apple pomace. As shown in Figure 9a,b, the reducing sugar yields of *Tt*GH74 hydrolysis were 13.88, 17.74 and 13.97 (RS g/kg DM) for sulfuric-acid-pretreated corn bran, DES-pretreated corn bran and DES-pretreated apple pomace, respectively, while the reducing sugar yields of GH10 xylanase hydrolysis were 16.54, 10.94 and 5.22, respectively, and the reducing sugar yields of CBH1/EG1 mixture hydrolysis were 81.41, 70.26 and 23.94, respectively. 

Correspondingly, when the enzymes act synergistically, the yields of reducing sugar by the combined action of GH10 xylanase with CBH1/EG1 mixture were 131.38, 96.70 and 27.72, respectively, and the degrees of synergy were 1.34, 1.19 and 0.95, respectively. The yields of the combined action of *Tt*GH74 with CBH1/EG1 mixture were 192.30, 119.84 and 47.08, and the degrees of synergy reached up to 2.02, 1.36 and 1.24, respectively. Furthermore, the yields of reducing sugar by the combined action of *Tt*GH74, GH10 xylanase and CBH1/EG1 mixture were 241.25, 152.83 and 77.19, and the degrees of synergy were 2.16, 1.54 and 1.79 for sulfuric-acid-pretreated corn bran, DES-pretreated corn bran and DES-pretreated apple pomace, respectively. 

The above results indicated that either *Tt*GH74 or GH10 xylanase showed a boosting effect on the hydrolysis efficiency of the CBH1/EG1 mixture, in comparison, the synergistic action between *Tt*GH74 and the CBH1/EG1 mixture was significantly higher than that of GH10 xylanase and the CBH1/EG1 mixture. Interestingly, the degrees of synergy between GH10 xylanase and *Tt*GH74 were 0.62, 0.88 and 0.69, respectively, indicating that GH10 xylanase and *Tt*GH74 not only had no synergy but also had an inhibition effect on each other due to the close spatial location of xyloglucan and xylan. However, the quaternary mixture of *Tt*GH74, GH10 xylanase and CBH1/EG1 resulted in more synergistic action than the ternary mixture of *Tt*GH74 with CBH1/EG1.

As shown in Figure 9a,b, for three pretreated substrates, when replacing *Tt*GH74 with *Tt*GH74ΔCBM in the enzymatic hydrolysis, the reducing sugar yields of *Tt*GH74ΔCBM hydrolysis were 9.54, 16.00 and 23.30 (RS g/kg DM) for sulfuric-acid-pretreated corn bran, DES-pretreated corn bran and DES-pretreated apple pomace, respectively. The yields of the combined action of *Tt*GH74ΔCBM with CBH1/EG1 mixture were 121.20, 105.39 and 65.06, and the degrees of synergy were 1.33, 1.22 and 1.38, respectively.

The yields of reducing sugar by the combined action of *Tt*GH74ΔCBM with GH10 xylanase and CBH1/EG1 mixture were 195.42, 145.68 and 98.80, and the degrees of synergy reached up to 1.82, 1.50 and 1.88, respectively. Similarly, no synergy but inhibition effect on each other was observed for *Tt*GH74ΔCBM and GH10 xylanase, and the degrees of synergy were 0.60, 0.82 and 0.44, respectively.

It can be inferred from the above results (Figure 9a,b) that the addition of *Tt*GH74 or *Tt*GH74ΔCBM into the CBH1/EG1 mixture or CBH1/EG1/GH10 mixture facilitated the overall hydrolysis of the three pretreated substrates. The boosting effects of *Tt*GH74 in the CBH1/EG1 mixture or CBH1/EG1/GH10 mixture was relatively higher than *Tt*GH74ΔCBM in terms of the degrees of synergy for sulfuric-acid-pretreated corn bran (*Tt*GH74 vs. *Tt*GH74ΔCBM, ternary and quaternary mixtures: 2.02 vs. 1.33 and 2.16 vs. 1.82) (*p* < 0.04) and DES-pretreated corn bran (1.36 vs. 1.22 and 1.54 vs. 1.50) (*p* < 0.05). 

In contrast, the boosting effects of *Tt*GH74ΔCBM was much higher than *Tt*GH74 in terms of the degrees of synergy for pretreated apple pomace (1.24 vs. 1.38 and 1.79 vs. 1.88) (*p* < 0.05). This is consistent with the bias of their activities when acted on three pretreated substrates alone (Figure 8), i.e., *Tt*GH74 showed higher activity towards pretreated corn bran than *Tt*GH74ΔCBM. Conversely, *Tt*GH74ΔCBM had much higher activity than *Tt*GH74 towards pretreated apple pomace.

In order to verify whether the above synergistic action was attributed to the presence of XG in the substrates, the associated XG was extracted by strong alkali treatment and XG-free residuals were used as substrates for synergistic experiments. As shown in Appendix A, tiny oligosaccharides could be released from strong alkali-treated residual by *Tt*GH74 hydrolysis based on the analysis of HPAEC-PAD, indicating that most of XG was removed by strong alkali treatment.

In addition, almost no reducing sugar was produced from the strong alkali-treated residual by GH10 xylanase hydrolysis (Figure 9c), indicating that the strong alkali treatment also removed most of the xylan from pretreated lignocelluloses. When XG and xylan-free substrates were hydrolyzed by the mixture of CBH1/EG1 and GH10 xylanase, CBH1/EG1 and *Tt*GH74 and the mixture of *Tt*GH74/CBH1/EG1/GH10 xylanase, no synergistic action was observed.

Their corresponding reducing sugar yields were almost the same as those of the CBH1/EG1 mixture alone, and all degrees of synergy were close to 1 (Figure 9c,d). However, we found that the hydrolysis efficiency of the CBH1/EG1 mixture on the strong alkali treated residues was greatly improved. The reducing sugar yields reached up to 192.29, 114.81 and 93.54 (RS g/kg DM), respectively, which were 136.20%, 63.41% and 290.72% (*p* < 0.01) higher than the corresponding reducing sugar yields of the CBH1/EG1 mixture on sulfuric-acid-pretreated corn bran, DES-pretreated corn bran and DES-pretreated apple pomace, respectively (Figure 9a). Thus, it further confirmed that the blocks of XG were one of the great significant factors impeding the degradation of xyloglucan-rich lignocellulose.

The structures, activity modes and enzymatic properties of GH74 xyloglucanases have been widely studied in the literature [17,18,41,42,43,44,45]; however, their roles in lignocellulose biomass degradation have been scarcely reported. Benko [30] et al. studied the contribution of the added GH74 xyloglucanase from *Trichoderma reesei* (*Tr*GH74) in the cellulase mixture to degradation of steam pretreated willow, barley straw, wheat straw, reed canary grass, corn stover and Solka Floc.They found that the degree of synergy of *Tr*GH74 with the cellulase mixture reached 1.22 for barley straw; however, only 1.07, 1.10 and 1.10 for corn stover, reed canary grass and willow, respectively. No synergistic effect was observed for wheat straw and Solka Floc.

Research suggested that the degree of synergy between xyloglucanase and the cellulase mixture was positively correlated with the ratio of xylose to glucose in the substrates. Overall, our study revealed that the degree of synergy between *Tt*GH74 and the cellulase mixture was much higher for both the pretreated corn bran or apple pomace when compared with the previous reports because of the higher content of XG in these residues. 

However, the influence factors involving synergy might be multiple and complicated. Apart from the XG content, the content of lignin and xylan and the structural modification of XG might also have a great influence on the synergy efficiency. The changes of composition and structure of XG in different pretreated residues in turn affected enzyme-substrate interaction. We observed that the degree of synergy between *Tt*GH74 and the cellulase mixture on sulfuric-acid-pretreated corn bran was much higher than that on DES-pretreated corn bran; however, they became close since the degrees of synergy were reduced by 34.16% and 10.29% on sulfuric-acid-pretreated and DES-pretreated corn bran when replacing with *Tt*GH74ΔCBM, respectively.

In view of the more and longer branched chains of XG in sulfuric-acid-pretreated compared with DES-pretreated corn bran (Figure 7b,d), it was reasonable to expect that CBM1 is conducive to *Tt*GH74 hydrolysis of the associated XG with more and longer branched chains in biomass. Thus, the yield of reducing sugars by *Tt*GH74 alone was 45.49% higher than *Tt*GH74ΔCBM on sulfuric-acid-pretreated corn bran but only 10.88% higher on DES-pretreated corn bran, leading to the degrees of synergy between *Tt*GH74 and cellulase mixture were 51.88% and 11.48% higher than that between *Tt*GH74ΔCBM and cellulase mixture on sulfuric-acid-pretreated and DES-pretreated corn bran, respectively (Figure 9a,b).

This further confirmed that the role and significance of CBM1 were substrate-specific because of differences in XG content and structure in various pretreated biomasses. 

Corn bran (or corn fiber) is an agricultural by-product obtained from corn processing. It is well-recognized that corn bran polysaccharides are very difficult to decompose through enzymatic hydrolysis [54,55,56,57]. The recalcitrance was regarded as due to the rich content of glucuronoarabinoxylan, which was extensively decorated with variations of both monomeric and oligomeric substitutions [58].

Apple pomace is the by-product from apple processing rich in pectin, cellulose and hemicelluloses [59], and it could be a raw material for biofuel and biochemical production [60]. However, the enzymatic saccharification of apple pomace has not been well investigated.

In this study, we demonstrated that both have rich in XG, which was closely associated with other polysaccharides and remained in solid residues after sulfuric acid or DES pretreatment. The associated XG significantly hindered the enzymatic hydrolysis efficiency. Our results demonstrated that the hydrolysis performance of cellulase mixture on pretreated XG-rich biomasses could be greatly improved by adding the so-called accessory enzyme xyloglucanase to enzyme mixture, suggesting the essential of xyloglucanase in enzyme cocktail for the efficient hydrolysis of XG-rich biomasses.

## 3. Materials and Methods

### 3.1. Materials

The chemicals and reagents used in this study were analytical grades. Xyloglucan (tamarind seed), β-glucan (barley), glucomannan (konjac), lichenan (lichen of Iceland), arabinoxylan (wheat) were purchased from Megazyme (Bray, Ireland); laminarin (laminaria), xylan (birch), pectin, carboxymethylcellulose sodium (CMC-Na) and Avicel were purchased from Sigma Aldrich (Shanghai, China); soluble starch (potato) and chitin were obtained from Sangon (Shanghai, China); and phosphoric acid-swollen cellulose (PASC) was prepared from Avicel according to the protocol described by Zhang [61] et al. Whatman filter paper was provided from Maidstone, UK.

Corn bran was collected from Nanyang, Henan Province, China. Destarched corn bran was prepared by amylase and papain treatment according to the method used by Rose and Inglett [62] with modifications. Amylase and papain were purchased from Imperial Jade Bio-Technology Co., Ltd. (Ningxia, China). Apple pomace was purchased from Yuanzhi Biotechnology Co., Ltd. (Shaanxi, China).

Endoglucanase 1 (EG1) from *Volvariella volvacea* and GH10 xylanase from *Eupenicillium parvum* 4–14 were expressed in *Pichia pastoris* KM71H according to previous methods [51,63], and cellobiohydrolase 1 (CBH1) from *Hypocrea jecorina* was purchased from Sigma-Aldrich (Shanghai, China).

### 3.2. Construction of TtGH74 and TtGH74ΔCBM

A gene fragment encoding mature xyloglucanase from *T. terrestris* NRRL 8126 (*Tt*GH74, XP_003650520.1) was synthesized by GenScript (Nanjing, China) with optimized codons. The codons were optimized using GenSmart™ Codon Optimization tool (Version Beta 1.0, https://www.genscript.com.cn/gensmart-free-gene-codon-optimization.html; accessed on 12 September 2019). The gene fragment was linked to the pPICZαA expression vector (Invitrogen, Carlsbad, CA, USA), located between the *Eco*RI and *Not*I sites. The CBM1 fragment in *Tt*GH74 was removed by PCR using the following primers (Table 4).

The sequences encoding signal peptide, linker peptide and CBM1 in selected genes were deleted, and only the catalytic domain (CD) was selected for sequence alignment.

### 3.3. Expression and Purification of TtGH74 and TtGH74ΔCBM

The recombinant plasmids were linearized by the restriction enzyme *Sac*I and integrated into the genome of *P. pastoris* KM71H host cells by electroporation. Transformed colonies were picked from yeast extract-peptone-dextrose medium containing 0.1% zeocin (YPDZ) to 100 mL buffered complex glycerol (BMGY) medium (100 mM potassium phosphate, pH 6.0, 1.34% (*w/v*) yeast nitrogen base (YNB), 1% yeast extract, 2% peptone and 1% glycerol) in a shaking incubator (200 rpm) at 28 °C until the culture reached an OD600 of 3.

The cells were harvested by centrifugation at 4000 rpm for 5 min and then decanted to 50 mL buffered methanol complex (BMMY) medium (100 mM potassium phosphate, pH 6.0, 1.34% (*w/v*) YNB, 1% yeast extract, 2% peptone). To induce expression, methanol was added to a final concentration of 1% every 24 h. After seven days, the culture was centrifuged (10,000 rpm, 10 min) and the supernatant was then directly loaded to the Ni-NTA pre-packed gravity column (Sangon Biotech, Shanghai, China). 

The purification steps were performed according to the manufacturer’s manual. The purified enzyme was dialyzed at 4 °C for 12 h to exclude imidazole. The protein concentration was quantified using a Pierce BCA Protein Assay kit (Thermo Fisher, Rockford, IL, USA). Polyacrylamide (12.5%) gel electrophoresis in 0.1% SDS was performed to determine the molecular mass of the enzyme preparation. Proteins were visualized after staining with Coomassie Brilliant Blue R-250.

The deglycosylation of the recombinant protein was performed by treating the protein with Endoglycosidase H (Endo H, endo-N-acetylglucosaminidase H of *Streptomyces plicatus*; NEB, Ipswich, MA, USA) according to the manufacturer’s instructions.

For optimizing the expression of *Tt*GH74, the effects of temperature (24 °C, 26 °C, 28 °C, 30 °C and 32 °C), methanol concentration (0.5% to 2.0%), pH (5.0, 6.0 and 7.0) and induction time (1 to 7 days) on the *Tt*GH74 expression were investigated step by step. For methanol concentration optimization, the pH was kept at 6.0, whereas for pH optimization, the methanol concentration was maintained at 1.5%, respectively. The culture temperature was 28 °C.

### 3.4. Properties of Recombinant TtGH74 and TtGH74ΔCBM from P. pastoris

The enzyme activity on tamarind seed XG was measured in a 0.5 mL reaction mixture containing 2 mg XG, 10 nM *Tt*GH74 or *Tt*GH74ΔCBM in 50 mM sodium acetate buffer (pH 5.5) at 75 °C for 15 min. The amount of reducing sugars released from the substrate was determined using the 3,5-dinitrosalicylic acid reagent (DNS) method [64]. One unit of enzyme activity was defined as the amount of enzyme that catalyzed the conversion of substrate to 1 μmol of glucose equivalent per minute. 

To determine the substrate specificity of *Tt*GH74, the following polysaccharides were selected: Tamarind seed XG, β-glucan, glucomannan, lichenan, arabinoxylan, laminarin, xylan, pectin, CMC-Na, soluble starch, chitin, Avicel and PASC. The reaction was performed at the same assay condition described above instead of the individually selected substrate. The end-products generated from tamarind seed XG by the *Tt*GH74 hydrolysis were analyzed using matrix assisted laser desorption/ionization time-of-flight mass spectrometry (MALDI-TOF MS).

The optimal temperatures and pHs, thermal and pH stability of *Tt*GH74 and *Tt*GH74ΔCBM were measured on tamarind seed XG. Optimum temperatures were determined at a range of temperature from 35 to 85 °C. Optimal pHs were determined in buffers in a range of pH 3.5–8.0 (50 mM sodium acetate buffer (pH 3.5–6.0), 50 mM sodium phosphate buffer (pH 6.0–8.0)). The maximum enzyme activity was taken as 100% and relative activities were calculated. 

The thermostability of the enzymes was determined by measuring the residual enzyme activities at assay condition after preincubating the recombinant enzymes at 50 to 70 °C for a specific time. The activity of *Tt*GH74 and *Tt*GH74ΔCBM before incubation was taken as 100% and the relative activities were calculated. The pH stability was tested by preincubating the recombinant enzyme in 50 mM buffer at various pH values (3.0–8.0) at 4 °C overnight, and the residual enzyme activities were measured as above. 

To investigate the effects of metal ions on *Tt*GH74 and *Tt*GH74ΔCBM activities, the enzyme activities were assayed at the presence of 1 mM or 5 mM metal ions (Li^+^, K^+^, Mg^2+^, Pb^2+^, Ca^2+^, Zn^2+^, Ba^2+^, Cu^2+^, Ni^2+^, Mn^2+^, Co^2+^, Fe^3+^, Al^3+^), NH^4+^ and EDTA, the reaction was conducted at 50 °C for 15 min. The *K*_m_ value against tamarind seed XG was determined at 50 °C for 7.5 min, using a substrate concentration in the ranges of 0.2–6 mg/mL. The *K*_m_ and other kinetic values were calculated with the GraphPad Prism 8.0 program.

### 3.5. Adsorption of TtGH74 and TtGH74ΔCBM on Different Substrates

Differences in adsorption of *Tt*GH74 and *Tt*GH74ΔCBM on different substrates were studied by incubating the *Tt*GH74 or *Tt*GH74ΔCBM in the mixture volume 0.5 mL with a final concentration of 300 μg/mL protein and 10 mg substrate in 50 mM sodium acetate buffer (pH 5.5) at 0 °C for 1 h with shaking (100 rpm). The substrates were Whatman filter paper, Avicel, PASC, sulfuric-acid-pretreated corn bran, DES-pretreated corn bran and DES-pretreated apple pomace. After the incubation, the mixture was centrifuged at 10,000 rpm for 10 min.

The unbound protein in the supernatant was quantified using a Pierce BCA Protein Assay kit. The adsorption of *Tt*GH74 on XG coated PASC was determined at the same incubation condition. The 0.5 mL mixture contained a final concentration of 300 μg/mL protein and 1 mg PASC with different proportions of XG. The XG coated PASC was prepared by premixing an aqueous solution of XG (0 to 600 (μg/mg)) with 1 mg PASC at room temperature in total 0.5 mL of 50 mM sodium acetate buffer (pH 5.5) with stirring (rotary disc contactor, 20 rpm) for 18 h [50]. The difference in binding affinity between *Tt*GH74 and *Tt*GH74ΔCBM to XG was also determined by affinity electrophoresis (AE). AE was conducted in 7.5% polyacrylamide gels containing 0.005 *w/v* soluble XG at 0 °C and 150 V for 3 h [65].

### 3.6. Hydrolysis Action of TtGH74 and TtGH74ΔCBM on XG and XG-coated PASC

The hydrolysis action of *Tt*GH74 or *Tt*GH74ΔCBM on free XG was performed in the reaction mixture 0.5 mL containing 10 nM *Tt*GH74 or *Tt*GH74ΔCBM and different amounts (25–600 μg) of XG in 50 mM sodium acetate buffer (pH 5.5) in the shaking incubator at 50 °C and 200 rpm for 1 h. Then the amount of reducing sugar was detected by the DNS method. The hydrolysis action of *Tt*GH74 or *Tt*GH74ΔCBM on XG-coated PASC was performed in the same reaction condition but using XG-coated PASC as substrate. 

The hydrolytic activity of EG1 or CBH1 towards XG-coated PASC was also performed in the same reaction condition but with 2.5 μg EG1 or 2.5 μg CBH1, respectively. In order to determine the presence of free XG in XG-coated PASC solution, the liquid fraction was separated by centrifugation and the unbound XG in the supernatant was extensively hydrolyzed by overdose *Tt*GH74, and the reducing sugar was measured by the DNS method.

### 3.7. Analysis of the Presence of XG and Hydrolysis Action of TtGH74 or TtGH74ΔCBM on Pretreated Residues

Corn bran and apple pomace were pretreated with two methods and different times. The destarched corn bran was pretreated either with 1% sulfuric acid or deep eutectic solvents (DES, lactic acid: ChCl molar ratio was 2:1). Apple pomace was also pretreated with DES as the same solution for destarched corn bran. The pretreatment temperature was 90 °C and reaction times were 3, 6, 9 and 12 h, respectively. 

After pretreatment, the residues were washed with distilled water to neutral pH and dried at 105 °C to a constant weight. In order to verify whether XG still exists in the pretreated substrates, the residues with different pretreatment times were hydrolyzed by *Tt*GH74 or *Tt*GH74ΔCBM in the 1 mL reaction mixture containing 20.7 nM of *Tt*GH74 (15 U/g substrate) or *Tt*GH74ΔCBM and 20 mg substrate with 0.1% antibiotics (zeocin and ampicillin) in 50 mM sodium acetate buffer (pH 5.5) in the shaking incubator at 50 °C and 200 rpm for 96 h. The reducing sugars produced after hydrolysis were detected by the DNS method. The presence of XG in the above three pretreated substrates was further confirmed by analyzing the *Tt*GH74 hydrolysis end-products using HPAEC-PAD and MALDI-TOF MS.

Finally, corn bran pretreated with sulfuric acid for 6 h, corn bran and apple pomace pretreated with DES for 9 h were used as substrates for further exploration. Time course of enzymatic hydrolysis of three pretreated substrates in different concentrations (from 10 to 80 mg) was performed by *Tt*GH74 or *Tt*GH74ΔCBM in the same condition described above. Samples were taken out every 12 h and the reducing sugars produced after hydrolysis were detected by the DNS method.

### 3.8. Synergistic Action of TtGH74 or TtGH74ΔCBM with CBH1/EG1 Mixture and Xylanase on Pretreated Lignocellulose

The synergy of *Tt*GH74 or *Tt*GH74ΔCBM with cellulase and xylanase on pretreated lignocelluloses was performed by mixed action of *Tt*GH74 or *Tt*GH74ΔCBM with CBH1/EG1 mixture, GH10 xylanase and GH10 xylanase/CBH1/EG1 mixture, respectively. The synergy of GH10 xylanase with CBH1/EG1 mixture on pretreated lignocellulose was also performed by mixed action GH10 xylanase with CBH1/EG1 in the same condition. 

The dosage of each enzyme was as follows: 20.7 nM of *Tt*GH74 or *Tt*GH74ΔCBM (15 U/g substrate), 15 U/g substrate of GH10 xylanase, 25 μg EG1 and 25 μg CBH1 for pretreated corn bran; while the dosage of each EG1 and CBH1 was reduced to 15 μg for DES-pretreated apple pomace because the cellulose content in DES-pretreated apple pomace is only about 60% of that in DES-pretreated corn bran. 

The synergistic action was conducted in a shaking incubator at 50 °C and 200 rpm for 96 h, and the volume of reaction mixture was 1 mL containing 20 mg substrate and appropriate enzymes in the 50 mM sodium acetate buffer (pH 5.5) with 0.1% antibiotics. The reducing sugars produced after hydrolysis were detected by the DNS method. The degree of synergy (DS) of the coupled enzyme mixture was calculated by Equation (1).
(1)$$\mathrm{DS}=\frac{{\mathrm{RS}}_{\mathrm{total}}}{{\mathrm{RS}}_{\mathrm{enzyme}\;1}{+\mathrm{RS}}_{\mathrm{enzyme}\;2}+{\mathrm{RS}}_{\mathrm{enzyme}\;3}+\ldots }$$
where RS_total_ is the reducing sugar released by enzymes when used together, and the denominator is the sum of the yield of reducing sugars when the enzymes are used separately in the same amounts as they were employed in the mixture.

In order to verify whether the synergistic action between xyloglucanase and other glycoside hydrolases was attributed to the presence of XG in the substrates, XG-free residues were prepared by extracting the associated XG by strong alkali treatment according to the literature [66]. Briefly, the pretreated materials were soaked in 15% NaOH at room temperature for 24 h. Then, the strong alkali treated residues were washed with distilled water to neutral pH and dried at 80 °C to a constant weight. 

Then, in order to verify whether XG was presented in the strong alkali treated residues, the residues were hydrolyzed by *Tt*GH74 and the released oligosaccharides were analyzed by HPAEC-PAD. The XG-free residues were used as substrates for synergistic experiments at the same reaction conditions as described above. The reducing sugars produced after hydrolysis were detected by the DNS method.

### 3.9. Test Method

The chemical composition analysis of pretreated solid fractions (lignin, glucose, xylose, galactose, mannose and arabinose) was performed according to the analytical procedure provided by the National Renewable Energy Laboratory (NREL/TP-510-42618 [67]), and the carbohydrates in supernatant were then quantified by high performance liquid chromatography (HPLC) (Agilent, Palo Alto, CA, USA). The HPLC system fitted with a Bio-Rad Aminex HPX-87H column was operated at 55 °C with 5 mM H_2_SO_4_ as the mobile phase at the flow rate of 0.6 mL/min as previously described by Chen et al. [68]. 

HPAEC-PAD analysis was performed on a Dionex ICS-5000 system (Dionex, Sunnyvale, CA, USA) equipped with pulsed amperometric detection (PAD) and a CarboPac PA200 analytical column (3 × 250 mm) with a CarboPac PA200 guard column (3 × 50 mm) according to Shi et al. described method [13]. MALDI-TOF MS analysis was performed on a 5800 MALDI TOF/TOF analyzer (AB SCIEX, Foster City, CA, USA) equipped with neodymium: yttrium-aluminum-garnet laser (laser wavelength was 349 nm) according to Shi et al. described method [13].

### 3.10. Statistical Analysis

All described experiments were performed in triplicate and all the reported data are means of the three samples. In addition, the Student’s *t*-test analysis was used to determine the statistical difference between the two groups, when *p* < 0.05, the data difference was statistically significant.

## 4. Conclusions

In this study, we characterized and comprehensively compared *Tt*GH74 and *Tt*GH74ΔCBM from *T. terrestris*. The deletion of CBM1 improved the thermostability but decreased the *K*_cat_ slightly and the sample almost completely lost its adsorption capacity for insoluble carbohydrates. We also compared the hydrolysis of free XG, XG-coated PASC, pretreated corn bran and apple pomace by *Tt*GH74 or *Tt*GH74ΔCBM action alone or in combination with other glycoside hydrolases. 

Our results indicated that XG was difficult to remove and still retained with some modifications in the residues of corn bran and apple pomace after common pretreatment. CBM1 might not be essential for the hydrolysis of free XG but it is effective for the associated XG to some extent. *Tt*GH74 could remove the bound xyloglucan on cellulose; therefore, it efficiently boosted the enzymatic hydrolysis of the pretreated XG-rich lignocellulosic biomasses. 

The presence of CBM1 is conducive to enzymatic hydrolysis; however, its role and significance are substrate-specific due to the differences in the XG content and structure in various pretreated biomasses. This study deepened our understanding of the effects of CBM1 on the enzymatic properties of GH74 xyloglucanases and its necessity in the degradation of various XG-rich biomasses.

## Figures and Tables

**Figure 1 ijms-23-05276-f001:**
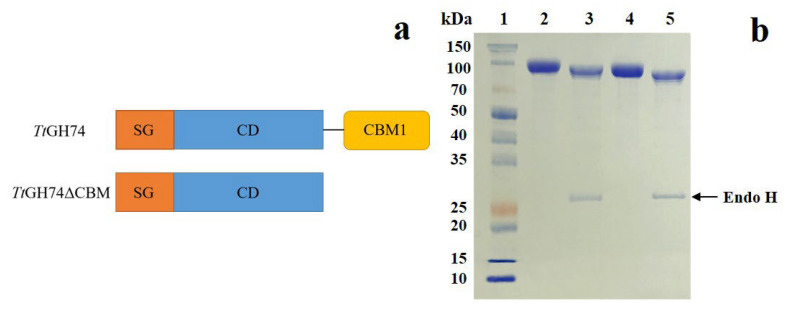
(**a**): The modularity of *Tt*GH74 and *Tt*GH74ΔCBM. SG, signal sequence; CD, catalytic domain module; CBM1, CBM1 module. (**b**): SDS-PAGE analysis of the purified *Tt*GH74 and *Tt*GH74ΔCBM and endo H-treated *Tt*GH74 and *Tt*GH74ΔCBM. Lane 1, molecular mass marker; Lane 2, *Tt*GH74; Lane 3, endo H-treated *Tt*GH74; Lane 4, *Tt*GH74ΔCBM; and Lane 5, endo H-treated *Tt*GH74ΔCBM.

**Figure 2 ijms-23-05276-f002:**
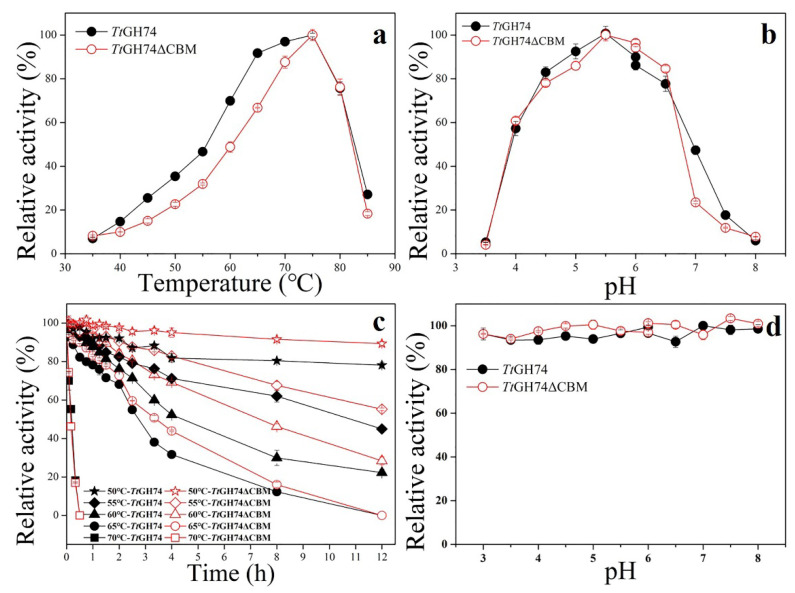
The effects of temperature (**a**) and pH (**b**) on the enzyme activity and the effects of temperature (**c**) and pH (**d**) on the stability of *Tt*GH74 and *Tt*GH74ΔCBM.

**Figure 3 ijms-23-05276-f003:**
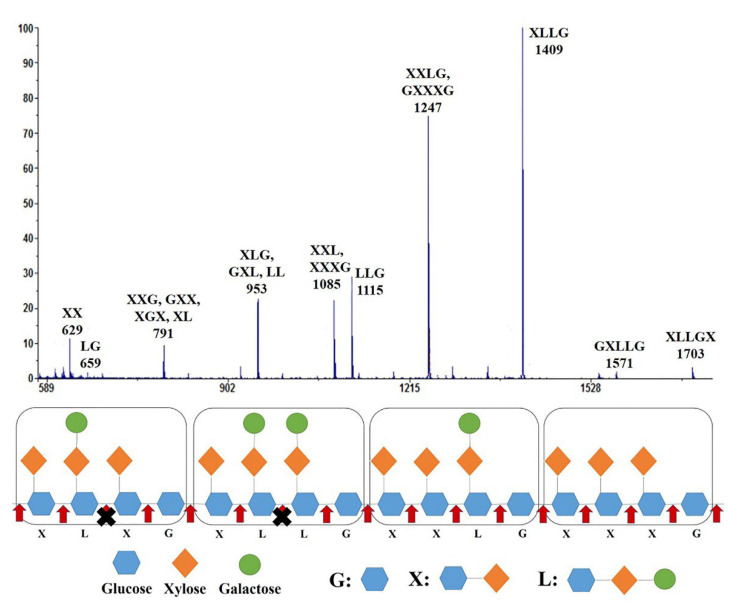
MALDI-TOF MS analysis of the products generated by *Tt*GH74 from tamarind seed XG. The red arrow shows the possible XG cleavage sites of *Tt*GH74, and the black cross indicates that *Tt*GH74 did not cleave at the reducing end side of L units in XLXG and XLLG.

**Figure 4 ijms-23-05276-f004:**
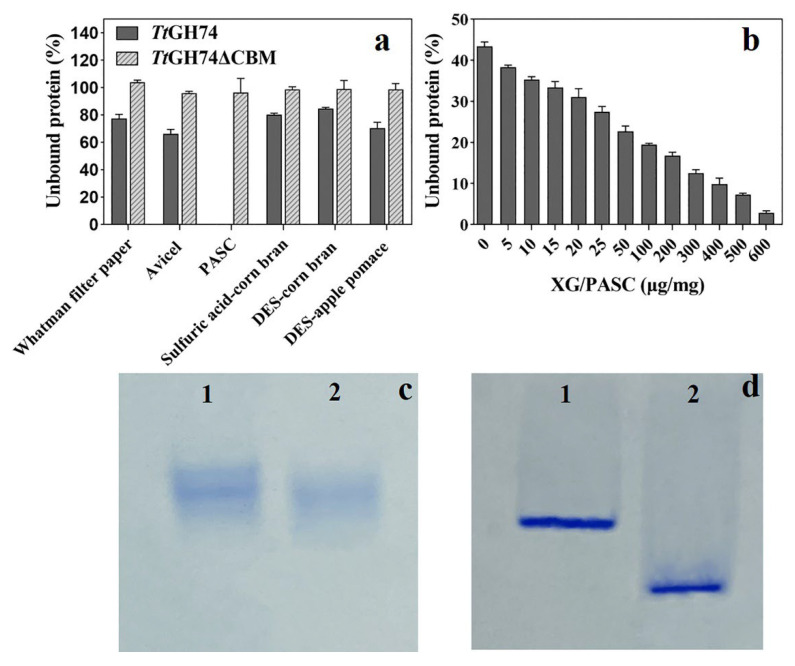
The adsorption differences of *Tt*GH74 and *Tt*GH74ΔCBM on various celluloses and lignocelluloses (**a**), the adsorption of *Tt*GH74 on cellulose with different proportions of XG (**b**) and analysis of their adsorption capacities on XG by electrophoresis in 7.5% native polyacrylamide gel without xyloglucan (**c**) or with 0.005 *w/v* xyloglucan (**d**). Lane 1, *Tt*GH74; and Lane 2, *Tt*GH74ΔCBM.

**Figure 5 ijms-23-05276-f005:**
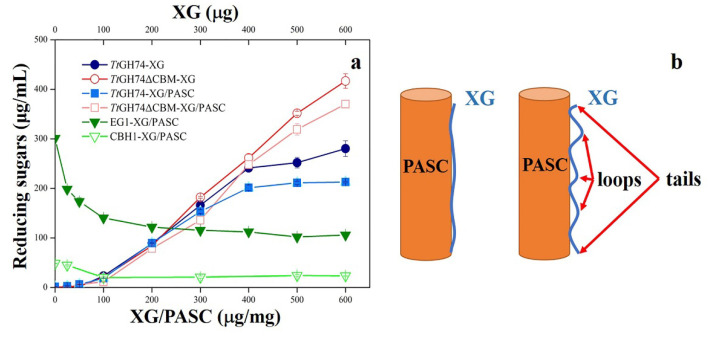
Hydrolysis of *Tt*GH74 and *Tt*GH74ΔCBM on different amounts of XG, *Tt*GH74, *Tt*GH74ΔCBM, EG1 and CBH1 hydrolyze PASC coated with different proportions of XG, respectively (**a**). The schematic diagram of the association pattern of XG and PASC (**b**). Orange cylinders represent PASC, and XG is represented by a blue line. The left diagram shows the low proportion of XG/PASC; the right diagram shows the high proportion of XG/PASC; and the XG forms accessible “loops” and “tails” on the PASC surface.

**Figure 6 ijms-23-05276-f006:**
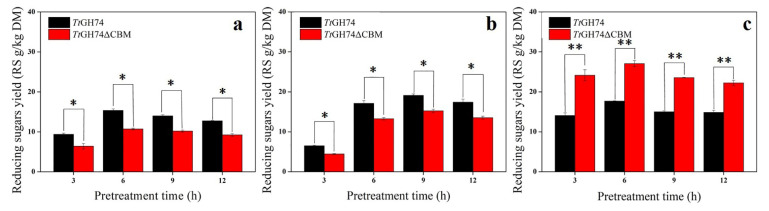
*Tt*GH74 and *Tt*GH74ΔCBM enzymatic hydrolysis of sulfuric-acid-pretreated corn bran (**a**), DES-pretreated corn bran (**b**) and DES-pretreated apple pomace (**c**) with different pretreatment times. The product yield was expressed in grams of reducing sugar produced by per kilogram of dry material (Reducing Sugars g/kg Dry Material, RS g/kg DM). The statistical differences between two groups of *Tt*GH74 and *Tt*GH74ΔCBM were determined using Student’s *t*-test analysis. Statistical significance is defined as **p* < 0.05 and ** *p* < 0.01.

**Figure 7 ijms-23-05276-f007:**
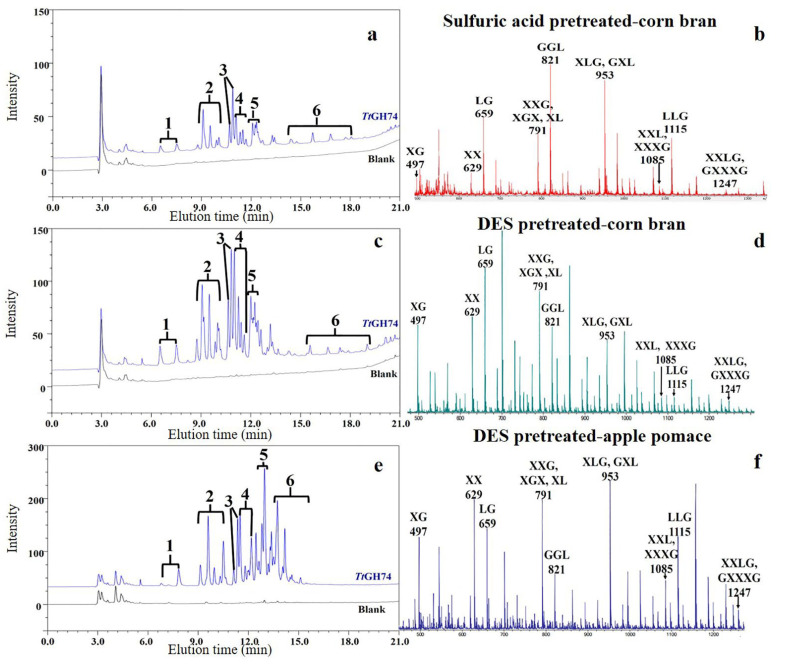
HPAEC-PAD (**left**) and MALDI-TOF MS (**right**) analysis of end-products generated by *Tt*GH74 from sulfuric-acid-pretreated corn bran (**a**,**b**), DES-pretreated corn bran (**c**,**d**) and apple pomace (**e**,**f**). The blank in the HPAEC-PAD represents a control experiment without *Tt*GH74.

**Figure 8 ijms-23-05276-f008:**
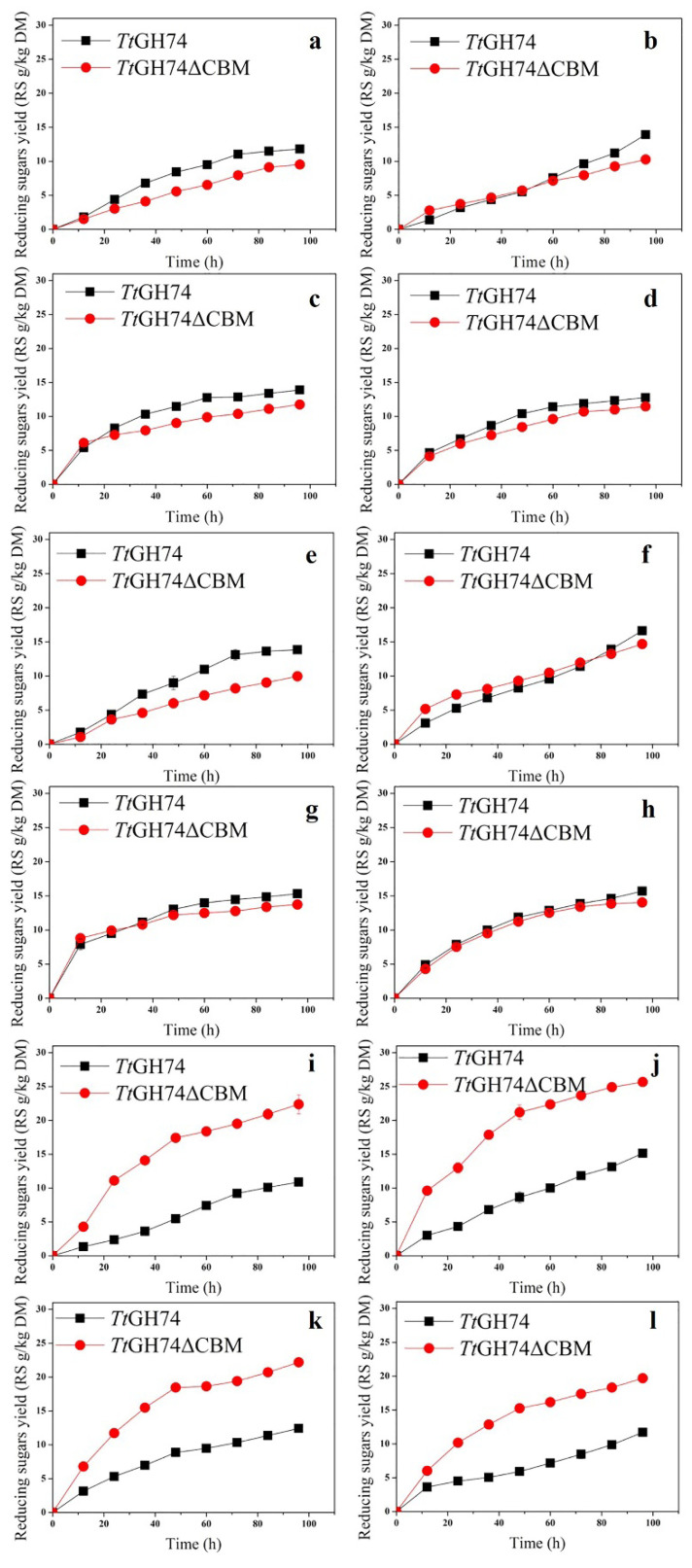
Time courses of the enzymatic hydrolysis of sulfuric-acid-pretreated corn bran, DES-pretreated corn bran and apple pomace by *Tt*GH74 or *Tt*GH74ΔCBM. (**a**–**d**): 10, 20, 40 and 80 mg of sulfuric-acid-pretreated corn bran, respectively; (**e**–**h**): 10, 20, 40 and 80 mg of DES-pretreated corn bran, respectively; (**i**–**l**): 10, 20, 40 and 80 mg of DES-pretreated apple pomace, respectively. The product yield was expressed in grams of reducing sugar produced by per kilogram of dry material (Reducing Sugars g/kg Dry Material, RS g/kg DM).

**Figure 9 ijms-23-05276-f009:**
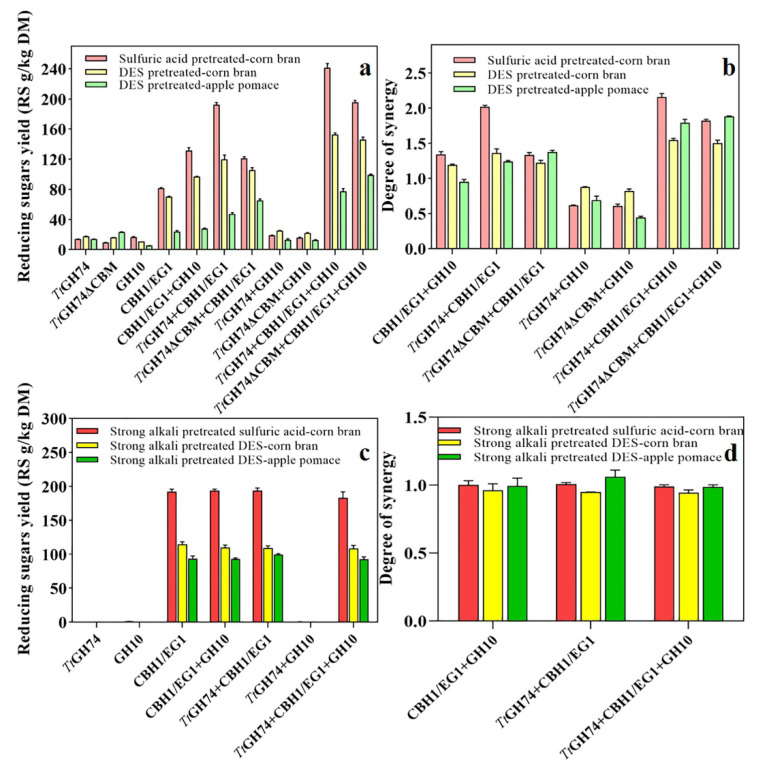
The synergy between *Tt*GH74, *Tt*GH74ΔCBM, GH10 xylanase and the CBH1/EG1 mixture on sulfuric-acid-pretreated corn bran and DES-pretreated lignocellulose (**a**,**b**). Enzymatic hydrolysis verification of pretreated lignocellulose components treated by strong alkali (**c**,**d**). The product yield was expressed in grams of reducing sugar produced by per kilogram of dry material (Reducing Sugars g/kg Dry Material, RS g/kg DM).

**Table 1 ijms-23-05276-t001:** The substrate specificity of *Tt*GH74.

Substrate	Glycosidic Bond	*Tt*GH74 Activity
β-Glucan (barley)	β-1,3 and β-1,4	++
Glucomannan (konjac)	β-1,4	++
Lichenan (lichen of iceland)	β-1,3 and β-1,4	+
Xyloglucan (tamarind seed)	β-1,4 α-1,6 and β-1,6	+++++
Laminarin (laminaria)	β-1,6-endo-β-1,3	
Soluble starch (potato)	α-1,4	
Xylan (birch)	β-1,4	
Arabinoxylan (wheat)	β-1,4	
Pectin	α-1,4	
Chitin	β-1,4	
CMC-Na	β-1,4	
Avicel	β-1,4	
PASC	β-1,4	+

+: 1%; ++: 5%; +++++: 100%.

**Table 2 ijms-23-05276-t002:** Comparison of the biochemical properties of *Tt*GH74 and *Tt*GH74ΔCBM with other GH74 xyloglucanases.

Entry Name	Strain	Temperature (°C)	pH	*V*_max_ (U/mg)	*K*_m_ (mg/mL)	*K*_cat_ (s^−1^)	References
*Tt*GH74	*Thielavia terrestris*	75	5.5	193.2	0.3225	283.36	This paper
*Tt*GH74ΔCBM				168.5	0.2671	233.09	
*Pc*GH74	*Phanerochaete chrysosporium*	55	6.0		0.25	28.1	[41]
*Pc*GH74ΔCBM					0.28	31.9	
*Mt*GH74	*Myceliophthora thermophila* VKPM	70–75	6.5		0.57		[42]
*Af*GH74	*Aspergillus fumigatus*	50	5.5	11.9	1.5	16.4	[43]
XEG74	*Paenibacillus sp* KM21	45	6.0	36.8	0.96	49.2	[44]
*Po*GH74	*Paenibacillus odorifer*	50	6.0		0.05	39.8	[45]

*K*cat: The number of moles of product that was catalyzed from substrate per mole of enzyme per min.

**Table 3 ijms-23-05276-t003:** The chemical composition of the pretreated corn bran and apple pomace substrates.

	Lignin	Glucose	Xylose	Galactose	Mannose	Arabinose
Sulfuric-acid-pretreated corn bran	20.73 ± 1.34%	66.04 ± 0.41%	8.10 ± 0.99%	0.18 ± 0.07%	1.82 ± 0.32%	0.74 ± 0.17%
DES-pretreated corn bran	11.10 ± 0.97%	70.04 ± 0.65%	7.78 ± 1.03%	0.09 ± 0.02%	0.98 ± 0.07%	0.43 ± 0.09%
DES-pretreated apple pomace	21.69 ± 1.83%	43.22 ± 0.32%	8.43 ± 1.32%	2.59 ± 0.15%	2.34 ± 0.18%	0.22 ± 0.06%

Data are the means of triplicate experiments with standard deviation (SD).

**Table 4 ijms-23-05276-t004:** Primers used in this study.

Primer	Sequence (5′ to 3′)
5′AOX	GACTGGTTCCAATTGACAAG
*Tt*GH74ΔCBM-R	ATAGTTTAGCGGCCGCTTAGTGATGGTGATGGTGATGGTGAGATTGAGTAGCTTGAGG

## Data Availability

The data presented in this study are available on request from the corresponding author.

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
