# Peer review of "Comparison of the Biochemical Properties and Roles in the Xyloglucan-Rich Biomass Degradation of a GH74 Xyloglucanase and Its CBM-Deleted Variant from Thielavia terrestris"

_ijms, 2022, doi:10.3390/ijms23095276_

Round 1
Reviewer 1 Report
This revision generally improve on the points I pointed out in the previous version, and I believe that the following corrections would be sufficient for publication in International Journal of Molecular Sciences.
Methods for statistical analysis should be described according to whether the comparison is between two groups or multiple groups (t-test, tukey HSD, Dunnett's test, etc.).
Alternatively, the Materials and Methods section could have a subsection on “Statistical analysis” and state, for example, "Student's t-test was used to compare paired samples, and Tukey HSD was used to compare multiple samples".
Author Response
Thank you, we added information about used statistical tests in new section 3.10 Statistical analysis.
Reviewer 2 Report
The responses to my comments (point by point), as well as corrections, are adequate.
Author Response
Thank you.
This manuscript is a resubmission of an earlier submission. The following is a list of the peer review reports and author responses from that submission.
Round 1
Reviewer 1 Report
TtGH74ΔCBM CBM deleted variants are equally good as native xyloglucanase TtGH74. Authors showed that CBM is not really important for the hydrolysis of free xyloglucan however may be effective for associated xyloglucan. It was also found that CBM deleted variant exhibited improved thermostability. Still, it was concluded that CBM is an important part of the enzyme.
Here I doubt in temperature stability results, as the results are only for 4 hours at temperature 50-70 degrees
Must do it atleast for 12 hours to say it is stable at elevated temperature.
Overall the results are exciting and huge amount of work. It is excellent.
Conclusions need to modify. To be more precise and …the last line of conclusions…’replace, ‘essential’ with, ‘necessity’.
Reviewer 2 Report
In this paper, Wang et al., purified the intact and the CBM-deleted recombinant proteins for GH74 xyloglucanase from Thielavia terrestris and analyzed their enzymatic properties. In addition, their hydrolytic activity was analyzed using XG and XG-coated PASC, and also the effect of solvent pretreatments on substrate degradation was evaluated using corn bran and apple pomace. A great number of experiments have been conducted and have provided some insights into the role of CBM and the increasing efficiency of XG degradation of biomasses, etc. However, the text is overall redundant and there a number of inconsistencies, some of which are noted below. Therefore, I recommend revising the overall manuscript structure and subsequent resubmission.
Individual points
Figure S2 does not appear in the main text. Also, Figure S3 is missing from the supplementary documents and is not mentioned in the text. Figure S4 appears before Figure S1, and the order of figures should be sequential. These are fatal error.
L114, The method or reference for codon optimization should be mentioned. This can be in Material and Methods.
Subsection 2.1.1. “Optimization of TtGH74 expression conditions” is not an essential part of this paper and the paper itself is redundant, so it should be listed together as 2.1 and compactly, not as a subsection.
In Table 1, there is a mention of PASC, but it should be spelled out if it is a first time PASC is mentioned in the text. Or, since there are many other abbreviations, they can be listed together in an Abbreviation section.
Do the ++ and + in Table 1 mean 5% and 1% activity, respectively, compared to the degradation activity of XG as a substrate? If so, it should be noted in a footnote in the table.
After L185, I did not immediately understand what X, G, and L meant. Are they Xylose, Glucose and Galactose, respectively? An explanation would be needed.
It would be more informative to point to Loops and tails in the cartoon in Fig. 5b.
L248, Data not shown is essentially unacceptable. It should be shown in the supplemental materials.
Figure 6 and etc., Is there a reason why there are no data for Sulfuric acid pretreated-apple pomace? Some explanation is needed.
What does the HPAEC-PAD chart mean? Does it show differences in the pattern of oligosaccharide formation due to different pretreatments? Explanation needed. Also, do you need the detailed numbers in the MALDI-TOF MS chart? The resolution is poor and the written numbers are not well discernible. If you don't need them, just erase them.
What are the blanks in the HPAEC-PAD analysis? I see small peaks on the blank as well, what are they derived from? Is it a gradient elution? Some explanations would be needed.
There is no statistical analyses on the data. For example, if you are discussing the results of a comparison between TtGH74 and TtGH74∆BMC, or the results of a comparison of the effects of three pretreatments, then proper statistical treatment would be necessary.